# Parser Evaluation for Analyzing Swedish 19th-20th Century Literature

**Sara Stymne[1], Carin Östman[2], and David Håkansson[2]**
[1]Department of Linguistics and Philology, Uppsala University, Sweden
[2]Department of Scandinavian Languages, Uppsala University, Sweden
sara.stymne@lingfil.uu.se
{carin.ostman,david.hakansson}@nordiska.uu.se

## Abstract

In this study, we aim to find a parser for accurately identifying different types of subordinate clauses, and related phenomena, in 19th–20th-century Swedish literature. Since no test set is available for parsing from this time period, we propose a lightweight annotation scheme for annotating a single relation of interest per sentence. We train a variety of parsers for Swedish and compare evaluations on standard modern test sets and our targeted test set. We find clear trends in which parser types perform best on the standard test sets, but that performance is considerably more varied on the targeted test set. We believe that our proposed annotation scheme can be useful for complementing standard evaluations, with a low annotation effort.

## 1 Introduction

Dependency parsers can be useful tools for analyzing large text materials, and as such can enable large-scale studies within many scientific disciplines. Modern parsers can achieve very high scores on standard test sets, at least for languages with large treebanks, but these test sets are often limited to only a few domains, and typically to publication-level modern language, such as news or Wikipedia. For more challenging text types, for instance, noisy data like Twitter or historical texts, parsers typically perform considerably worse even for high-resource languages.

Parsers are typically evaluated on a treebank that is split into training, development, and test sets. This can overestimate the parser performance, since parsers are then trained on data that matches its test set in all relevant aspects, such as genre, time period, and annotation style. Furthermore, parser evaluation is typically done using metrics that give a holistic score for the full tree, such as (un)labeled attachment score. In many real-world scenarios, such as ours, we are not interested in the full tree, but in a subset of relations.

This study is part of a larger project with the overall aim to identify and explore language change in Swedish literature during the period 1800–1930. During the 19th century, the Swedish language changed in several aspects. This change includes various linguistic levels and also involves lexical aspects. Overall, the changes led to a smaller difference between spoken and written Swedish since the written language moved closer to the spoken vernacular (see Section 3). The goal of the project is to cover morphological, syntactical, and lexical changes. In this paper, however, we focus only on syntactic aspects, focusing on subordinate clauses. The changes in the 19th century resulted in a less complex language — not least as far as subordinate clauses and related phenomena are concerned. To enable large-scale analysis of subordinate clauses, we require a high-quality parser for our target domain, Swedish literary novels and short stories from 1800–1930. In this paper, we explore whether parsers can be evaluated for this domain, without requiring a large manual annotation effort.

To evaluate a parser for a new text type and task, as in our case 19th century literature with a focus mainly on subordinate clauses, we would ideally like to have an annotated treebank for the target text type. However, this is a human annotation task that is time-consuming, and thus costly, and which requires an expert on dependency grammar. For many practical projects, this is not feasible. We propose a lightweight annotation task for our target task, which consists of only annotating one type of phenomenon per sentence. The focus is on four phenomena related to subordinate clauses, for which we annotate a small targeted test set for our target text type. For comparison, we also evaluate

| Relation | Example | Translation | Class |
|---|---|---|---|
| CLEFT | Det var här han skulle *anfallas* . | 'It was here that he would be *attacked* .' | Correct |
| CLEFT | Det skola vi *göra* klockan åtta . | 'That we should *do* at eight o'clock' | Wrong |
| RELCL | Hvad hon beundrar Mauritz , som kan *stå* så lugn ! | 'How she admires Mauritz , who can *stand* so calmly !' | Correct |
| RELCL | Men kan du säga hvar vi *äro* ? | 'But can you tell me where we *are* ?' | Wrong |
| CCOMP | Se till att du inte *halkar* . | 'Make sure that you do not *slip* .' | Correct |
| CCOMP | Må den aldrig mer *komma* för mina ögon ! | 'May it never again *come* before my eyes !' | Wrong |
| NO-AUX | Jag har fått hvad du i natt *skrifvit* till mig . | 'I have received what you [have] *written* for me tonight .' | Correct |
| NO-AUX | Enhälligt ha vi *kommit* fram till detta slut : | 'Unanimously , we have *reached* this end :' | Wrong |

Table 1: Examples of sentences shown to the annotators, marked as either correct or wrong.

on standard Swedish test sets. Table 1 shows examples of each class, where the task is to identify if a given word is the head of a specific subordinate clause type or if it is a clausal complement without the auxiliary 'have'.

We compare several variants of three generations of parsers trained on different subsets of the Universal Dependencies (UD) treebanks (Nivre et al., 2020), and evaluate them on UD, both with holistic metrics and for a subset of relations of interest, as well as on our targeted test set. On the UD test sets we see clear trends that a modern transformer-based parser is better than BiLSTM- and SVM-based parsers, and that it is better to train on several North Germanic languages than only on Swedish. However, on our new targeted test set, the results are more mixed, and we see less clear trends, which is in line with earlier work for German (Adelmann et al., 2018). We think that our targeted test set is able to give a complementary view to standard evaluations, but that the sampling procedure can be improved.

In Section 2 we review related work, followed by a description of our project focused on Swedish language change in Section 3. In Section 4 we describe the data and in Section 5 we describe the parsers evaluated, including the multilingual training setup. We present the results in Section 6, discuss them in Section 7, and finally, we conclude in Section 8.

## 2 Related Work

Dependency parsers have continuously developed, from 'old school' parsers like MaltParser (Nivre et al., 2007) and MSTparser (McDonald et al., 2005) based on classical machine learning, like support vector machines, to modern neural parsers. Many of the first strong neural parsers were based on recurrent neural networks, as most of the best parsers in the CoNLL 2017 shared

task on dependency parsing (Zeman et al., 2017). Next, models based on deep contextualized embeddings have been taking over, and most strong parsers today are based on fine-tuning contextualized models like BERT (Devlin et al., 2019) or XLM-R (Conneau et al., 2020), e.g. Machamp (van der Goot et al., 2021) and Trankit (Nguyen et al., 2021).

The standard way to evaluate dependency parsers is by calculating holistic metrics such as labeled attachment score (LAS), which measures the percentage of words which gets both their head word and label correct. There are, however, examples of more detailed evaluations (e.g. McDonald and Nivre, 2007; Kulmizev et al., 2019; Salomoni, 2017), focusing on aspects such as arc and sentence lengths, non-projective dependencies, and scores for specific POS-tags and dependency relations. The overall conclusion is typically that different parser types have different strengths, e.g. that graph-based parsers tend to perform better than transition-based parsers on long-distance dependencies (McDonald and Nivre, 2007). As far as we are aware, there are no datasets and evaluations like our proposal, focused on a single relation per sentence.

Highly relevant to our study is the work of Adelmann et al. (2018), who evaluate a set of six parsers for digital humanities research, focusing on German novels and academic texts. Like us, they are also interested in specific relations, for instance, related to speaker attribution, and not only in holistic evaluation. Unlike us, they perform a full dependency tree annotation effort for three sample texts. In addition, they do not include any neural parsers in their evaluation. They find that several parsers do well on the holistic metrics, but that the results are considerably worse for several of the specific relations of interest, such as appositions, and that it is not always the overall strongest

parser that is the best choice for a specific relation. Salomoni (2017) performed a detailed evaluation on parsing German 17th-century literature, for which he annotated two excerpts of text with full dependency annotations. Again, no neural parsers were included in the study, which found a drop compared to in-domain results, but where the relative performance of the two parsers evaluated was consistent on different metrics, possibly because of the large difference in performance between them.

Swedish literary texts from different eras have been analyzed for different purposes before, requiring taggers and/or parsers. Dahllöf (2022) aims to characterize differences between dialogue and narrative in contemporary fiction, whereas Stymne et al. (2018b) analyze prose rhythm in a novel from 1940. However, in none of these studies, the choice of tagger and/or parser is motivated. There have also been some earlier smaller-scale studies focusing on the transition towards a more colloquial written Swedish. For instance, language development in Swedish literature during the 19th century has been explored, but only on a small scale focusing on individual authors (e.g. Lindstedt, 1922; Von Hofsten, 1935).

## 3 Language Change in 19th Century Swedish

This study is part of a larger project with the overall aim to identify and explore language change in Swedish literature during the period 1800–1930. In the history of the Swedish language, this period is characterized by modernization in the sense that the written language was influenced by the spoken vernacular. In this process of modernization, fictional prose is of certain interest since it has been suggested that linguistic change spread from literary dialogue (Engdahl, 1962; Teleman, 2003). By investigating a corpus of literary texts the project will not only contribute with a more detailed account of language change in 19th-century Swedish but also address the question of how linguistic change increased in the community.

The modernization of the Swedish written language during the 19th century affected several linguistic aspects. As for the lexicon, it is well-known that formal function words were replaced by colloquial counterparts. Much attention has also been devoted to the loss of verbal agreement, i.e. the use of the vernacular singular variant in both singular and plural. On the syntactic level, Engdahl (1962) has shown a remarkable change in sentence length during the end of the 19th century. Engdahl's study focuses on non-fictional prose, periodicals from 1878 to 1950, but his results call for a more detailed account of syntactic complexity during the period, and hence we will focus on subordinate clauses and phenomena related to them in this paper.

For this study, we have chosen to focus on three types of subordinate clauses, based on UD dependency labels, and one phenomenon related to subordinate clauses: (i) *relative clauses* (RELCL), (ii) *cleft constructions* (CLEFT),[1] (iii) *clausal complements* not determined by obligatory control (CCOMP), and (iv) *auxiliary drop* (NO-AUX). Whereas the first three types can be used in order to measure syntactic complexity, auxiliary drop has been suggested to mark written style, and hence almost never occur in spoken language (cf. Wellander, 1939). Since auxiliary drop of finite verbs is restricted to subordinate clauses in Swedish, we have included it as related to subordinate clauses. In this study, we only include auxiliary drop that occurs in clausal complements, CCOMP. Examples of the selected clause types are shown in Table 1.

## 4 Data

In this section, we will describe the existing data from UD and the new targeted dataset we constructed for this project

### 4.1 Universal Dependencies Treebanks

We use data from Universal Dependencies (Nivre et al., 2020) version 2.11 (Zeman et al., 2022) for training our parsers and for the standard evaluation. Besides dependency annotations, UD also contains lemmas, universal and language-specific part-of-speech tags (UPOS/XPOS), and morphological features. Our main focus is on Swedish, for which there are three treebanks, Talbanken, LinES, and PUD, where PUD only contains a test set. In addition, we use data from related North Germanic languages: Norwegian (both variants: Bokmål and Nynorsk), Danish, Faroese, and Icelandic. The treebanks used are summarized in Table 2. The intuition behind also using related lan-

---

[1] In UD, both relative clauses and cleft constructions are subtypes of ACL, clausal modifier of noun, and are denoted ACL:RELCL and ACL:CLEFT. In this paper, we will use shorter names, excluding the prefix.

| Language | Treebank | Code | Genres | Train | Test |
|---|---|---|---|---|---|
| | Talbanken | sv_t | news, nonfiction | 67K | 20K |
| Swedish | PUD | sv_p | news, wiki | – | 19K |
| | LinES-M | sv_lm | fiction, nonfiction, spoken | 18K | 73K |
| | Bokmaal | no_b | blog, news, nonfiction | 244K | 30K |
| Norwegian | Nynorsk | no_n | blog, news, nonfiction | 245K | 25K |
| | NynorskLIA | no_nl | spoken | 35K | 10K |
| Danish | DDT | da | fiction, news, nonfiction, spoken | 80K | 10K |
| Faroese | FarPaHC | fo | bible | 1.5K | 6.6K |
| Icelandic | Modern | is | news, nonfiction | 7.5K | 10K |

Table 2: Treebanks used, with info about genres (as defined in UD) and the number of tokens in test and training data. LinES-M refers to our modified version of LinES.

guages is twofold, first, it has been shown to improve parsers (e.g. Smith et al., 2018a), second, we believe it may make the parser more robust to non-standard Swedish, which has many differences from the modern Swedish of the Swedish treebanks. Written Norwegian and Danish, in particular, are very similar to Swedish, and are considered mutually intelligible.

As can be seen in Table 2, the genres, according to the UD specification, of the treebanks used are mixed. To be able to, at least some extent, investigate whether it would help to have an in-genre test set, we create a modified version, LinES-M, of the LinES treebank (Ahrenberg, 2007) which consists of three genres: literary fiction, Microsoft manuals, and European parliament proceedings. The literary part contains a set of novels translated from English, published 1977–2017. While this is not a perfect match to our target of novels and short stories written originally in Swedish during an earlier time period, this was the closest we could get to an in-domain test set, without any re-annotation. We re-split LinES by merging the data from the training and test sets, and moving all literature[2] to a new test set, and all other texts to a new training set, referred to as LinES-M in Table 2.

For evaluation on the UD test sets, we report labeled attachment score (LAS). For LinES-M, we also report F1-scores for the three relations in focus for our targeted test set and AUX, which is relevant for identifying auxiliary drop.

## 4.2 Targeted Literature Dataset

In this section, we will describe the sampling and annotation of the targeted literary dataset annotated for this project as an alternative way of

evaluating the performance of parsers on specific phenomena in a specific text type. The targeted dataset is publicly available under the Creative Commons license, CC BY-NC-SA.[3]

### 4.2.1 Text Selection and Processing

Our target data is literary texts from 1800–1930, focusing on novels and collections of short stories. Such works have been made available by Litteraturbanken.[4] We choose to work only with the subset of works that have been proofread after going through OCR, available in an XML format. We extracted all novels and short stories available in this format from the time period of interest. From these texts, we extracted the raw text paragraphs, and tokenized the text. For another sub-project, we had already extracted a set of novels where quotation marks are used to mark dialogue, and used the quotation marks to separate dialogue and narrative, which we use also in this study. This sample consists of 165 novels and collections of short stories. The data was parsed early on in the project, using Swepipe and UUparser[s] with Swepipe tags (see Section 5).

The annotation task was designed to be simple and fast. Thus we decided to focus on a single relation of interest per sentence. From the parse trees, we extracted all sentences containing an arc labeled with a relation of interest and marked the modifier of the arc, which is the headword of the specific subordinate clause.[5] Figure 1 shows a parsed example sentence, containing a relative clause, with an arc from the headword *Mauritz* to the modifier *stå* ('stand'), which is the head of the

---

[2]The literary works are in documents 2,3,4,6,7, and 8; document 1 contains Microsoft manuals and document 5 contains parliament proceedings (Lars Ahrenberg, personal communication).

[3]https://github.com/UppsalaNLP/SweSubEval

[4]https://litteraturbanken.se/

[5]It would also be possible to consider other more complex annotations, such as also including the head of the relevant arc, to ensure that the subordinate clause is attached at the correct position, or to require that the span of the subordinate clause is correct.

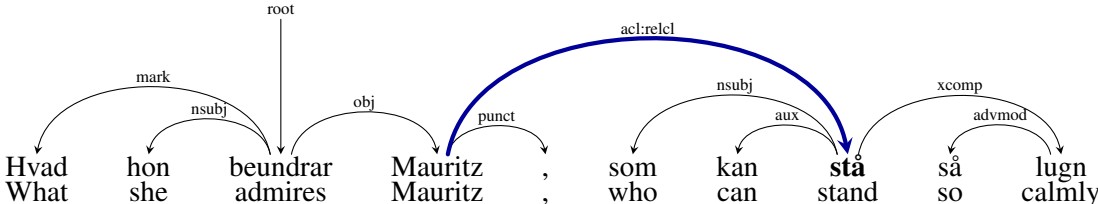

Figure 1: Parsed dependency tree (UUParser) for the sentence *Hvad hon beundrar Mauritz, som kan *stå* så lugn* 'How she admires Mauritz, who can *stand* so calmly', with English glosses added. The arc of interest, ACL:RELCL with 'stand' as a modifier, is marked in blue.

relative clause *som kan stå så lugn* ('who can stand so calmly'), where the marked word thus would be *stå*, as an instance of the type RELCL. For NO-AUX, we also checked that there was no outgoing AUX relation from the marked word. It is not uncommon to have several instances of a single relation type in a sentence, but we only marked a single occurrence per example, to make the annotation consistent between sentences. From this set, we randomly sampled 200 sentences for each relation type, except CLEFT, for which we only found 74 examples, which were all included. Table 1 shows annotated examples of each class, where we also see examples of old plural verb forms like *äro* (modern: *är*, 'are') and old-fashioned spelling like *'skrifvit'* (modern: *skrivit*, 'written').

### 4.2.2 Annotation

The annotation was performed by the last two authors, both native Swedish speakers, and researchers in Scandinavian languages with expertise in Swedish grammar. The annotators were given the example sentences in Excel, and for each sentence, they were to decide whether the marked head word belonged to the given type or not. For each type, 20 examples were annotated by both annotators, and the remaining examples were split between them. After the first round, there were a few disagreements in the doubly annotated sets, which were discussed by the annotators, followed by a re-annotation of all examples. The initial round of annotation was very quick, roughly between 15–30 minutes per 100 examples, with a somewhat longer time needed for CCOMP. Table 3 shows the number of correct and wrong examples for each class. Note that the dataset is skewed towards positive examples.

### 4.2.3 Evaluation

We evaluate on the targeted dataset by calculating the number of times the parser assigns the cor-

| Relation | Correct | Wrong |
|---|---|---|
| CLEFT | 64 | 10 |
| RELCL | 133 | 67 |
| CCOMP | 141 | 59 |
| NO-AUX | 170 | 30 |

Table 3: Class distribution in our annotated dataset

rect relation to the focus word, and for NO-AUX, that there in addition is no aux-dependent. We then calculate precision and recall for each relation type. Note that recall may be overestimated by this procedure since we do not cover any examples not identified by a parser. This evaluation is different from standard evaluation of dependency parsers where we evaluate a full tree. In this case, we instead evaluate a single relation of interest for each sentence.

## 5 Parsers

In order to investigate how well the different types of evaluation work, we explore three generations of parsers. As a baseline, we use the easily accessible Swepipe with its provided model for Swedish. We also use two generations of neural parsers, UUParser and Machamp, for which we also experiment with multilingual parsing. We train each model three times with different random seeds and report average scores.

### 5.1 Swepipe

As a baseline parser, we wanted an easily accessible parser, which comes with a trained parsing model, and which might be used by non-experts in a digital humanities project. Our choice was to use the Swedish annotation pipeline, Swepipe.[6], a pre-trained model covering all steps needed to analyze Swedish texts from scratch, including tokenization, tagging, and parsing. Swepipe is similar

---

[6]https://github.com/robertostling/efselab

to several other systems targeted at this user group, such as the web-based Swegram,[7] which uses the same parser and tagger (Megyesi et al., 2019).

Swepipe is pre-neural and uses efselab (Östling, 2018) for tagging and MaltParser (Nivre et al., 2007) trained on Talbanken for parsing. Malt-Parser is a classical transition-based parser, using a support vector machine for classification, based on a feature vector with words, POS-tags, and already built relations.

## 5.2 UUParser

UUParser (de Lhoneux et al., 2017; Smith et al., 2018b) is a neural transition-based dependency parser with a BiLSTM feature extractor, based on Kiperwasser and Goldberg (2016). Word representations are fed to a BiLSTM, to create contextualized word representations, which are given as input to an MLP classifying the next transition. We use an arc-hybrid transition model (Kuhlmann et al., 2011) with a swap transition (Nivre, 2009) and a static-dynamic oracle (de Lhoneux et al., 2017). As input word representation we use word embeddings, character-based word embeddings, UPOS-tag embeddings, and treebank embeddings, which represent the treebank of a sentence. All embeddings were initialized randomly at training time. When applying UUparser on new texts, we need a proxy treebank that indicates which of the treebanks from training for use as the treebank embedding at test time, for which we always use Talbanken, since it is present in all models, and it performed well in Stymne et al. (2018a). We use the default UUparser settings (Smith et al., 2018b), except for adding drop-out with a rate of 0.33 for UPOS-embeddings, since the parser is trained with gold tags. At test time, we use two different sets of POS-tags, from Swepipe/efselab and from Machamp. We will call these variants UUparser[s] and UUparser[m] respectively. To counteract the differing sizes of the training data, we limited the number of sentences used per treebank to 4,300 per epoch.

## 5.3 Machamp

Machamp (van der Goot et al., 2021) is a toolkit for multitask learning covering several NLP tasks, based on fine-tuning a pre-trained contextualized model, like BERT (Devlin et al., 2019). In a multi-task setup, each task has a separate decoder. The

| Group | Included treebanks/languages |
|---|---|
| Talbank | Swedish-Talbanken |
| Swedish | Talbank+ Swedish-LinES-M |
| SweNor | Swedish + Norwegian (*3) |
| Scand | SweNor + Danish |
| NorthG | Scand + Faroese + Icelandic |

Table 4: Groups of languages/treebanks used for multilingual training. See Table 2 for specific treebanks.

dependency parser is a graph-based parser using deep biaffine attention (Dozat and Manning, 2018) to score word pairs, and the CLU algorithm (Chu and Liu, 1965; Edmonds, 1967) to extract trees. For tagging, a greedy decoder, with a softmax output layer is used.

In this work we use Machamp in a multi-task setup, to jointly learn tagging of UPOS, XPOS, and morphological features, and dependency parsing. We experiment with two sets of language models, multilingual BERT (mBERT Devlin et al., 2019),[8] trained on 104 languages including all languages used in our study except Faroese, and the Swedish model KB-BERT (Malmsten et al., 2020), trained only on Swedish. We will call these systems Machamp[m] and Machamp[k] respectively. For both models, we used the cased version.[9] KB-BERT has been shown to improve Swedish named entity recognition and POS-tagging (Malmsten et al., 2020), but as far as we are aware, it has not been used in multilingual dependency parsing models. We use the default parameters of Machamp. To counteract the differing sizes of the training data, we applied sampling smoothing set to 0.5.

## 5.4 Multilingual Training

For UUParser and Machamp, we explore multilingual training. We limit ourselves to the North-Germanic languages, all relatively closely related to Swedish. We train two Swedish models, on Talbanken only, to be comparable with Swepipe, and also with LinES-M. In addition, we train three models with different subsets of the other North Germanic languages. For our multilingual models, we first combine Swedish with Norwegian, which has three treebanks covering both variants

---

[7]https://cl.lingfil.uu.se/swegram/

[8]https://github.com/google-research/bert/blob/master/multilingual.md

[9]We used models from HuggingFace (https://huggingface.co/models), for KB-BERT: KB/bert-base-swedish-cased and for mBERT: bert-base-multilingual-cased.

|  | LAS | | | F1, LinES-M | | | |
|---|---|---|---|---|---|---|---|
|  | LinES-M | TB | PUD | CLEFT | RELCL | CCOMP | AUX |
| Swepipe-Talbank | 71.75 | 79.69 | 78.82 | – | 61.31 | 54.98 | 88.45 |
| UUparser$^m$-Talbank | 72.10 | 83.75 | 76.66 | 26.82 | 64.67 | 59.62 | 93.99 |
| UUparser$^m$-Swedish | 75.51 | 83.76 | 77.50 | 29.12 | 67.37 | 61.65 | 94.21 |
| UUparser$^m$-Norswe | 79.69 | 85.60 | 81.50 | 39.92 | 74.34 | 66.79 | 94.35 |
| UUparser$^m$-Scand | 79.74 | 85.43 | 81.34 | 41.74 | 73.03 | 64.93 | 94.20 |
| UUparser$^m$-NorthG | 79.33 | 85.35 | 81.27 | 41.71 | 72.82 | 64.70 | 94.27 |
| Machamp$^k$-Talbank | 80.54 | **92.24** | 86.05 | **56.73** | 79.07 | **74.59** | 95.44 |
| Machamp$^k$-Swedish | 80.26 | 90.72 | **86.83** | 49.67 | 75.84 | 71.29 | 93.94 |
| Machamp$^k$-Norswe | **83.13** | 91.63 | 86.79 | 55.42 | **81.29** | 75.32 | 95.29 |
| Machamp$^k$-Scand | **83.16** | **92.31** | 87.21 | 55.54 | 81.21 | 74.27 | **95.97** |
| Machamp$^k$-NorthG | **83.03** | **92.35** | 87.17 | **56.00** | 82.27 | 74.78 | 95.85 |

Table 5: Results on standard Swedish UD test sets. LAS for all three Swedish test sets, and F1-scores for four relations of interest for LinES-M.

|  | Precision | | | | Recall | | | |
|---|---|---|---|---|---|---|---|---|
|  | CLEFT | RELCL | CCOMP | NO-AUX | CLEFT | RELCL | CCOMP | NO-AUX |
| Swepipe-Talbank | – | 66.33 | 70.41 | 84.62 | 0.00 | **99.25** | **98.57** | **97.06** |
| UUparser$^m$-Talbank | 92.46 | 93.32 | 94.11 | 98.14 | 50.35 | 82.37 | 63.97 | 51.44 |
| UUparser$^m$-Swedish | 92.49 | 93.45 | 95.84 | 97.60 | 69.79 | 81.45 | 65.95 | 50.85 |
| UUparser$^m$-NorSwe | 92.12 | 94.65 | **97.39** | 98.30 | **84.55** | 81.20 | 70.87 | 56.21 |
| UUparser$^m$-Scand | 94.64 | **95.69** | **96.73** | 98.72 | **84.20** | 79.62 | 70.48 | 61.05 |
| UUparser$^m$-NorthG | 93.31 | 95.55 | **96.06** | **99.05** | 75.00 | 79.37 | 74.13 | 61.57 |
| Machamp$^k$-Talbank | 94.12 | 95.16 | 94.63 | 98.52 | 59.90 | 83.46 | **75.48** | **65.69** |
| Machamp$^k$-Swedish | 94.92 | **96.19** | 95.09 | 98.81 | 53.12 | 82.21 | **73.81** | 65.10 |
| Machamp$^k$-NorSwe | 95.38 | **96.71** | 94.77 | **99.13** | 72.92 | 79.70 | 73.33 | **67.25** |
| Machamp$^k$-Scand | **96.61** | 95.11 | 94.29 | **99.01** | 59.38 | **87.47** | 66.90 | 58.82 |
| Machamp$^k$-NorthG | **95.38** | 93.83 | 93.46 | 99.00 | 64.06 | **87.72** | 68.10 | 58.04 |

Table 6: Precision and recall for our targeted test set.

of Norwegian. We then add Danish, to train a Scandinavian model. The reason for adding Norwegian first, despite the fact that Danish is considered a closer relative to Swedish, is the availability of more data for Norwegian with variability in language variants. Our final model, NorthG, also adds Faroese and Icelandic, which are more distant from Swedish, and not mutually intelligible. The language groups are summarized in Table 4.

## 6 Results

Tables 5 and 6 show results from the standard and targeted evaluations for Swepipe, UUparser$^m$ with Machamp$^k$ POS-tags and Machamp$^k$ trained with KB-BERT. In all tables, we mark the three best results for each metric in bold. While our focus is on Swedish, which is reported in this section, we also report results with Machamp for the additional languages used for training our parsing models in Appendix A.

Table 5 shows results on UD test sets. We see no obvious differences between the LAS performance pattern on the in-genre LinES-M and the other two Swedish test sets, indicating that genre may not play a big role in this case; contempo-

rary novels are likely relatively close to the news, non-fiction, and wiki texts in the other Swedish treebanks. Swepipe has overall the lowest scores, followed by UUparser$^m$, and then Machamp$^k$. For the two Swedish models, the differences between using only Talbanken and adding the small LinES-M training set are typically small, but sometimes with a positive effect for UUparser$^m$ and a negative effect for Machamp$^k$.[10] Adding Norwegian leads to improvements in nearly all scores, often quite substantial, whereas adding additional languages has a smaller impact. The difference between parsers varies for the different relation types. Swepipe does not find any CLEFTs, and falls behind UUparser$^m$ on all other relation types, especially for AUX. Machamp$^k$ improves considerably over UUparser$^m$ for all explored relations, except AUX, where both neural parsers perform well, possibly since they both use the POS-tags of Machamp$^k$.

---

[10]This may be due to the fact that in Machamp, treebanks are simply concatenated, but in UUparser, they are distinguished by treebank embeddings, which has been shown to improve results when training on different treebanks for the same language (Stymne et al., 2018a). We leave an investigation of this issue to future work.

| | LAS | | | F1, UD_LinES-M | | | | P, litt | | | |
|---|---|---|---|---|---|---|---|---|---|---|---|
| | LinES-M | TB | PUD | CLEFT | RELCL | CCOMP | AUX | CLEFT | RELCL | CCOMP | NO-AUX |
| Swepipe-Talbank | 71.75 | 79.69 | 78.82 | – | 61.31 | 54.98 | 88.45 | – | 79.52 | 82.14 | 90.41 |
| UUparser$^s$-Talbank | 70.80 | 82.35 | 75.78 | 26.08 | 63.01 | 58.39 | 91.31 | 92.80 | 92.52 | 93.05 | 96.50 |
| UUparser$^s$-Scand | 77.63 | 83.39 | 80.25 | 30.77 | 70.55 | 62.22 | 90.82 | 93.86 | 94.07 | **94.66** | 97.95 |
| UUparser$^m$-Talbank | 72.10 | 83.75 | 76.66 | 26.82 | 64.67 | 59.62 | **93.99** | 92.46 | 93.32 | 94.11 | 98.14 |
| UUparser$^m$-Scand | 79.74 | 85.43 | 81.34 | 41.74 | 73.03 | 64.93 | 94.20 | **94.64** | **95.69** | **96.73** | 98.72 |
| Machamp$^m$-Talbank | 77.20 | 89.35 | 84.21 | 38.47 | 72.87 | 69.09 | 92.91 | 92.94 | **96.13** | 93.00 | 98.23 |
| Machamp$^m$-Scand | **80.13** | 89.50 | 85.79 | 43.09 | 77.67 | **71.18** | 93.49 | 93.41 | **96.98** | 92.47 | **99.08** |
| Machamp$^k$-Talbank | 80.54 | 92.24 | 86.05 | **56.73** | 79.07 | **74.59** | 95.44 | 94.12 | 95.16 | 94.63 | 98.52 |
| Machamp$^k$-Scand | **83.16** | 92.31 | 87.21 | 55.54 | 81.21 | 74.27 | 95.97 | 96.61 | 95.11 | 94.29 | **99.01** |

Table 7: Comparison of parser variants, on standard test sets and our test set.

The results in Table 6 for our targeted test set show a partially different picture. First, we note that Swepipe has a very high recall for all relation types except CLEFT, which it never predicts. We think this is mainly an artifact of the sampling procedure for this test set, where the annotated sentences were sampled from Swepipe and UUparser$^s$, with Swepipe POS-tags, which means that they were mostly predicted as correct by Swepipe. The other parsers do not have this advantage and thus have a lower recall, which we believe is more predictive of real performance, even though it still may be overestimated due to the sampling procedure. Swepipe has considerably lower precision than the other parsers for all relation types. We believe that the evaluation should still be fair in comparing UUparser$^m$ and Machamp$^k$, from which no samples were taken. Compared to the standard evaluation where Machamp$^k$ was clearly better than UUparser$^m$, we now see a more mixed picture, where there is no clear overall advantage of Machamp$^k$ over UUparser$^m$, and the results are mixed across relation types and precision/recall. The trends between training languages are also less clear, with some combinations standing out in performance for some relation types. Machamp$^k$ trained with Scand and NorthG has a considerably higher recall on RELCL than the other models, with only a small drop in precision. On CCOMP and NO-AUX, on the other hand, these two models instead have a low recall, without gaining much on precision. We do not see this pattern for UUparser$^m$, where the Scand model is overall strong.

In Table 7 we show a summary of results for both variants of UUparser and Machamp, showing only precision for the targeted test set, since recall is biased towards Swepipe and UUparser$^s$ due to the sampling.[11] We can see that UUparser$^s$ does not consistently improve on LAS over Swepipe when trained on the same Talbanken data, but that adding the Scandinavian treebanks improves the results considerably both for the UD evaluations and on the targeted test set. When we compare the two variants of UUparser and Machamp we see that UUparser$^m$ and Machamp$^k$ beat their variant consistently on the UD evaluation, and in most cases on the targeted test set. We also see that training on Scand is better than training on Talbanken in the majority of cases, both for UD and on precision for the targeted test set, however, from Table 6, we know that Scand is sometimes not as strong on recall.

## 7 Discussion

An important question is whether the parser performance on our target task is good enough to use for our study of change in the Swedish written language. Overall, both Machamp and UUparser have good precision for all our relations of interest, always scoring above 90, and reaching scores above 96 for some parsers for each relation type. The recall, however, is considerably lower. This means that the instances of each relation type the parser finds are mostly good, but it does miss a substantial part of relevant instances, especially given the fact that all examples are sampled from a parser, and we might have missed additional instances. The recall is highest for RELCL, where it is well above 80 for several of the models both for Machamp and UUparser. This approaches a level that is usable for our end project, of finding syntactic features in 18th–19th-century literature, and tracking them over time. Other relation types have a more mixed performance, as CLEFT, for which UUparser$^m$ trained on NorSwe and Scand performs very well, with a recall of over

---

[11]To save space, we only show results for two training

language groups. The other groups exhibit largely the same trends.

84, but where other models perform considerably worse. The recall of CCOMP, and especially of NO-AUX is lower, and we would need to improve parser performance for those relation types, possibly by using domain adaptation techniques, before they would reach a useful level. The varying performance of parsers for different relation types is in line with the results for German by Adelmann et al. (2018), who recommend choosing different parsers for different end goals.

On the standard evaluation, Machamp is clearly overall better than UUparser, training on Scand is better than training only on Swedish, KB-BERT is better than mBERT for Machamp, and UUparser is better with Machamp tags than with Swepipe tags. For our targeted test sets, however, we see fewer clear trends, and there is much more variation among the systems. Machamp$^k$ and UUparser$^m$ tend to perform better than their counterparts, and the multilingual models may have a small advantage over the Swedish-only models. Swepipe clearly seems to fall behind the other parsers on precision, whereas its high recall can be explained by the sampling procedure. A side-effect of our study is that we have found that Machamp$^k$ trained on Scand or NorthG is a very strong parser for modern Swedish as measured by the UD test sets.

Our targeted test set does suffer from an issue with sampling from only two parsers, which affects its recall mainly for Swepipe, but also for UUparser$^s$. We believe UUparser$^m$ is less affected since it relies on a different set of POS-tags. The dataset is also relatively small, especially for the CLEFT relation. However, we think it still contributes to showing that when selecting a parser for a particular target task and text type, we cannot rely solely on evaluation scores on standard test sets, as also shown by Adelmann et al. (2018). Even if we focus on the F1-score for the relations of interest in Lines-M, rather than on the full tree, we see no clear similarity of parser ranking to the evaluation of the same relation types in our targeted test set. To further investigate whether this type of test set can indeed be useful, we would need to perform further analysis. It would be interesting to learn more about where the main improvements shown on UD evaluation for a parser like Machamp$^k$ actually occurs. We also think it would be useful to consider the sampling for the test set, specifically to also annotate some raw text, in order to find out what type of instances are not identified by any of our parsers. Another issue that we did not yet explore, is whether parsing performance varies over the time period in question.

## 8  Conclusion

We describe a study of Swedish dependency parsers with the goal of tracking changes in the use of certain types of subordinate clauses and related phenomena in Swedish literature from 1800–1930. Since standard test sets do not cover this time period or genre, and we did not have the resources to perform a full annotation of dependency trees, we propose a smaller-scale annotation task, focusing on single relation types. We evaluated a set of parsers on UD and on our targeted test set. While there was a clear and relatively consistent order between the parsers on the UD evaluation, the performance was more mixed on our targeted test set, without a clear overall best parser across relation types. We believe that our proposed annotation scheme can be useful in complementing standard evaluations, with a low annotation effort, but that more analysis is needed.

## Acknowledgments

This work is funded by the Swedish research council under project 2020-02617: *Fictional prose and language change. The role of colloquialization in the history of Swedish 1830–1930*. We would like to thank Johan Svedjedal and Joakim Nivre for their helpful discussions about this work, and the anonymous reviewers for their insightful comments. Computations were enabled by resources in project UPPMAX 2020/2-2 at the Uppsala Multidisciplinary Center for Advanced Computational Science.

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

## A Results for Additional Languages

Table 8 shows results for all treebanks used during training when parsed with Machamp either fine-tuned on top of the Swedish KB-BERT model (Malmsten et al., 2020), or the multilingual mBERT model (Devlin et al., 2019), when trained on the different language groups (see Table 4. We note that for Danish and Norwegian, which are very closely related to Swedish, models including these languages in the parsing training data perform nearly as well when trained on

|  |  | sv_t | sv_lm | sv_p | no_b | no_n | no_nl | da | fo | is |
|---|---|---|---|---|---|---|---|---|---|---|
| Talbank | KB-BERT | 92.24 | 80.54 | 86.05 | *72.33* | *64.33* | *45.62* | *62.61* | *18.56* | *11.00* |
|  | mBERT | 89.35 | 77.20 | 84.21 | *77.93* | *75.10* | *51.89* | *67.73* | *37.15* | *48.31* |
| Swedish | KB-BERT | 90.72 | 80.26 | 86.83 | *71.78* | *63.84* | *46.03* | *62.14* | *17.03* | *9.76* |
|  | mBERT | 86.43 | 76.77 | 84.21 | *77.82* | *74.68* | *51.87* | *67.71* | *35.53* | *46.62* |
| SweNor | KB-BERT | 91.63 | 83.13 | 86.79 | 91.51 | 90.91 | 75.01 | *69.15* | *20.81* | *13.85* |
|  | mBERT | 89.56 | 79.82 | 85.68 | 92.28 | 91.77 | 75.98 | *72.25* | *37.08* | *50.38* |
| Scand | KB-BERT | 92.31 | 83.16 | 87.21 | 91.73 | 91.36 | 75.57 | 86.44 | *21.42* | *14.50* |
|  | mBERT | 89.50 | 80.13 | 85.79 | 92.01 | 91.63 | 75.41 | 87.13 | *38.36* | *49.62* |
| NorthG | KB-BERT | 92.35 | 83.03 | 87.17 | 91.94 | 91.47 | 75.68 | 86.54 | 63.49 | 31.94 |
|  | mBERT | 89.49 | 79.99 | 85.77 | 92.15 | 91.49 | 75.94 | 87.24 | 71.82 | 60.30 |

Table 8: Results for all languages used for training models with Machamp fine-tuned based on Swedish KB-BERT or multilingual mBERT. Codes for treebanks refer to Table 2. Scores marked in italics indicate languages that were not present in the parser training data. All languages except Faroese are present in the mBERT pre-training data.

top of the Swedish KB-BERT model as on top of the mBERT model trained on 104 languages including Norwegian and Danish. It thus seems that for very similar languages a strong language model for a close language is just as good as a multilingual model containing many unrelated languages. However, this only holds when Danish and Norwegian are among the languages in the parsing training data; when the parser is trained only on Swedish, it is better to use mBERT than KB-BERT. Icelandic and Faroese are less closely related to Swedish than Danish and Norwegian, and for these languages, it is always better to use mBERT than KB-BERT. It is also notable that the performance is much poorer than for Danish and Norwegian. Faroese, which is not present in mBERT, performs quite poorly both with KB-BERT and mBERT when not present in the parser training data, but quite well with both models when present in the training data, whereas Icelandic even in that case performs poorly with KB-BERT. Overall, we see that Machamp with mBERT trained with the NorthG model is a strong parser for all the included languages. However, adding Icelandic and Faroese to the Scandinavian model has only a minor impact on the Scandinavian languages.