# OpenReview forum: "Parser Evaluation for Analyzing Swedish 19th-20th Century Literature"
_NoDaLiDa/2023/Conference — NoDaLiDa 2023_

### Official Review · Reviewer_Vokc · 2023-03-10
**A technically competent evaluation of parser performance in the context of a specific cross-disciplinary research project, with nevertheless some unclarities in the purpose and design of the evaluation.**

**Rating:** 8
**Confidence:** 4

**Review:**

The authors present a technically competent evaluation of parser performance, focussing on specific relations rather than the standard full tree evaluation. This focus is motivated by the intended use of these parsers in a historical linguistic research project. As the authors point out, these alternative views of parser performance may be different from the standardly reported performance, and are thus valuable when it comes to applying a parser in a specific task. This is a method that is worth promoting.

I found a couple of unclarities in presentation that are easily fixed (listed below), but I also have a more serious, more profound question about the value of used test set and the conclusions that may be drawn about the parser performance on the specific task the authors set out to evaluate (also in the list below). The latter point makes the paper weaker in my opinion, but it doesn't take away the more general point that looking at task specific aspects of performance is a path worth pursuing.

- In the introduction there are statements like
 >Overall, the changes led to a smaller difference between spoken and written Swedish since the written language moved closer to the spoken vernacular

  and
  > The changes in the 19th century resulted in a less complex language — not least as far as subordinate clauses and related phenomena are concerned

  This is picked up later again in Section 3, with some references, but here it would be good if the authors could include a pointer to that section so that it is clear that these claims to not stand without back-up.

- "formal function**s** words" → "formal function words"
- End of Section 3, I think quick examples of the chosen relations/phenomena would be good.
- Table 2 needs at least translations
- Having read until Section 4.2 I don't really understand the point of the annotation. It appears only the relation label is annotated on the dependant. As a relation is between two things, it would make sense to mark at least the head and the dependent? Or, if you are looking at these things as heads in a subordinate clause, that is, the question really asked to the annotators is: is this the main verb in a relative clause? is this the main verb in a subordinate clause with auxiliary drop? etc, that would also implicitly assume you have the concept of clause available, which presupposes a subtree or a constituent. Neither are available here.  What do you do with this single bit of information? (Perhaps it comes later in the paper but I'm jotting this comment down now because the purpose should be made more clear earlier.)
- (Leaving the previous comment up because I think it is a possible point of improvement) Having read the whole paper I understand the evaluation procedure better. In the context of the research questions that the use of these parsers will help answer, it would make sense to also consider two other bits of information: how well do these parsers "bracket" / identifiy the whole subordinate close, how well do they indentify the nominal/verbal head of the relative/complement clauses.
- The way the test set is constructed measures precision, right? You extract tokens/sentences with a certain property attributed to them by the parsers used in the preprocessing step, and then the annotators check whether this is correctly attributed. This measures precision of the extraction procedure. There is an unknown number of missed clefts, relative clauses and dropped auxes in the material. When you later on measure recall of the parsers against these annotations, one must be careful not to interpret this as recall of the parsers on the material on a whole, but rather as a ceiling value for this recall. So even when you say that
 > [T]he annotated sentences were sampled from Swepipe and UUparsers, with Swepipe POS-tags, which means that they were mostly predicted as correct by Swepipe. The other parsers do not have this advantage, and thus have a lower recall, which we believe is more predictive of real performance.

  I think that the truth is that even for these other parsers, recall on the test set __by design__ is going to be a (considerably) over-optimistic view of real performance.
- "randomaly" → "randomly" (l510)


**Paper Type:**

Long paper

---

### Official Review · Reviewer_JhLf · 2023-03-10
**An evaluation of parsers on specific types of subordinate clauses in Swedish literature from 1800-1930.**

**Rating:** 8
**Confidence:** 3

**Review:**

The paper describes an evaluation of three different parsers on a set of clauses that have been identified problematic in text from 1800-1930.

Strengths: Well structured evaluation, that is clearly presented. The insights gained from the results of the study are valuable for the NLP community.

Weaknesses: There is no mention of how the different parsers perform on modern, unproblematic clauses. No examples of problematic clauses from the data are provided.



**Paper Type:**

Long paper

---

### Decision · Program_Chairs · 2023-03-17

Accept